# Application of Deep Learning for Delineation of Visible Cadastral Boundaries from Remote Sensing Imagery

**Sophie Crommelinck \***, **Mila Koeva**, **Michael Ying Yang** and **George Vosselman**

Faculty of Geo-Information Science and Earth Observation (ITC), University of Twente, Hengelosestraat 99, 7514 AE Enschede, The Netherlands; m.n.koeva@utwente.nl (M.K.); michael.yang@utwente.nl (M.Y.Y.); george.vosselman@utwente.nl (G.V.)

**\*** Correspondence: s.crommelinck@utwente.nl; Tel.: +31-53-489-5524

**Abstract:** Cadastral boundaries are often demarcated by objects that are visible in remote sensing imagery. Indirect surveying relies on the delineation of visible parcel boundaries from such images. Despite advances in automated detection and localization of objects from images, indirect surveying is rarely automated and relies on manual on-screen delineation. We have previously introduced a boundary delineation workflow, comprising image segmentation, boundary classification and interactive delineation that we applied on Unmanned Aerial Vehicle (UAV) data to delineate roads. In this study, we improve each of these steps. For image segmentation, we remove the need to reduce the image resolution and we limit over-segmentation by reducing the number of segment lines by 80% through filtering. For boundary classification, we show how Convolutional Neural Networks (CNN) can be used for boundary line classification, thereby eliminating the previous need for Random Forest (RF) feature generation and thus achieving 71% accuracy. For interactive delineation, we develop additional and more intuitive delineation functionalities that cover more application cases. We test our approach on more varied and larger data sets by applying it to UAV and aerial imagery of 0.02–0.25 m resolution from Kenya, Rwanda and Ethiopia. We show that it is more effective in terms of clicks and time compared to manual delineation for parcels surrounded by visible boundaries. Strongest advantages are obtained for rural scenes delineated from aerial imagery, where the delineation effort per parcel requires 38% less time and 80% fewer clicks compared to manual delineation.

**Keywords:** cadastral mapping; indirect surveying; boundary extraction; boundary delineation; machine learning; deep learning; image analysis; CNN; RF

---

## 1. Introduction

Recording land rights provides land owners tenure security, a sustainable livelihood and increases financial opportunities. Estimates suggest that about 75% of the world population does not have access to a formal system to register and safeguard their land rights [1]. This lack of recorded land rights increases insecure land tenure and fosters existence-threatening conflicts, particularly in developing countries. Recording land rights spatially, i.e., cadastral mapping, is considered the most expensive part of a land administration system [2]. Recent developments in technology allow us to rethink contemporary cadastral mapping. The aim of this study is to make use of technological developments to create automated and thus more efficient approaches for cadastral mapping.

Automated cadastral boundary delineation based on remote sensing data has been rarely investigated, even though physical objects, which can be extracted using image analysis, often define visible cadastral boundaries [3].



Visible cadastral boundaries demarcated by physical objects such as hedges, fences, walls, bushes, roads, or rivers, can make up a large portion of all cadastral boundaries [4,5]. Indirect surveying relies on the delineation of such visible parcel boundaries from remote sensing imagery.

Automatically extracting visible cadastral boundaries combined with (legal) adjudication and an incorporation of local knowledge from human operators offers the potential to improve current cadastral mapping approaches in terms of time, cost, accuracy and acceptance [6]. High-resolution remote sensing imagery, increasingly captured with Unmanned Aerial Vehicles (UAVs) in land administration [7–11], provides the basis for such a semi-automated delineation of visible boundaries.

### 1.1. CNN Deep Learning for Cadastral Mapping

CNNs are one of the most popular and successful deep networks for image interpretation tasks. They are proven to work efficiently to identify various objects in remote sensing imagery [12–16]. Comprehensive overviews contextualizing the evolution of deep learning and CNNs in geoscience and remote sensing are provided by Bergen et al. and Zhu et al. [17,18]. In essence, CNNs are neural networks that incorporate the convolution and pooling operation as a layer. CNNs have been characterized by five concepts [19]:

- Convolution operation increases the network's simplicity, which makes training more efficient.
- Representation learning through filters requires the user to engineer the architecture rather than the features.
- Location invariance through pooling layers allows filters to detect features dissociated from their location.
- Hierarchy of layers allows the learning of abstract concepts based on simpler concepts.
- Feature extraction and classification are included in training, which eliminates the traditional machine learning need for hand-crafted features, and distinguishes CNN as a deep learning approach.

In deep learning, there are two approaches to train a CNN: From scratch or via transfer learning [20]. When trained from scratch, all features are learned from data to be provided, which demands large amounts of data and comes with a higher risk of overfitting. An over-fitted network can make accurate predictions for a certain dataset, but fails to generalize its learning capacity for another dataset. With transfer learning, part of the features are learned from a different, typically large dataset. These low-level features are more general and abstract. The network has proven excellence for a specific application. Its core architecture is kept and applied to a new application. Only the last convolution block is trained on specific data of the new application, resulting in specialized high-level features. Transfer learning requires learning fewer features, and thus fewer data. In our study, we investigate transfer learning as an existing CNN for cadastral mapping.

### 1.2. Study Objective

The main goal of our research is to develop an approach that simplifies image-based cadastral mapping to support the automated mapping of land tenure. We pursue this goal by developing an automated cadastral boundary delineation approach applicable to remote sensing imagery. In this study, we describe our approach in detail, optimize its components, apply it to more varied and larger remote sensing imagery from Kenya, Rwanda and Ethiopia, test its applicability to cadastral mapping, and assess its effectiveness compared to manual delineation.

We previously proposed a semi-automated indirect surveying approach for cadastral mapping from remote sensing data. To delineate roads from UAV imagery with that workflow, the number of clicks per 100 m compared to manual delineation was reduced by up to 86%, while obtaining a similar localization quality [21]. The workflow consists of: (i) Image segmentation to extract visible object outlines, (ii) boundary classification to predict boundary likelihoods for extracted segment lines, and (iii) interactive delineation to connect these lines based on the predicted boundary likelihood. In this study, we investigate improvements in all three steps (Figure 1). First, for step (i), we filter out

small segments to reduce over-segmentation. Second, for step (ii), we replace hand-crafted features and line classification based on Random Forest (RF) by Convolutional Neural Networks (CNNs).

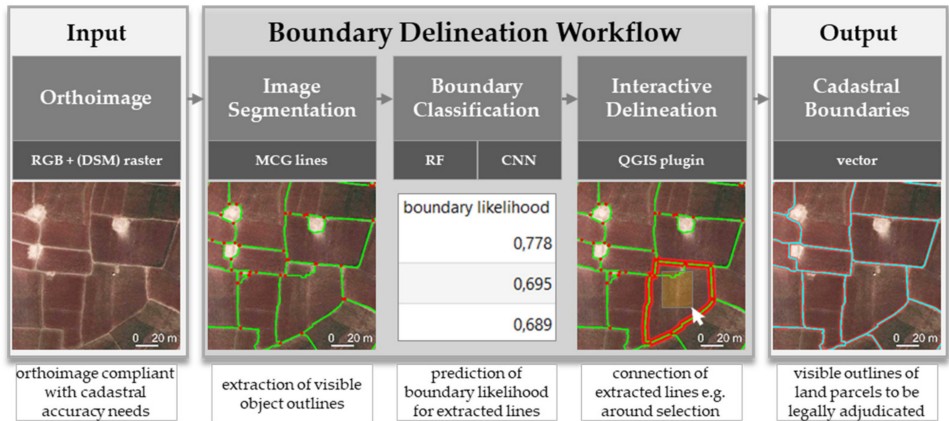

**Figure 1.** Boundary Delineation workflow proposed to improve indirect surveying. This study optimizes image segmentation, questions whether Random Forest (RF) or Convolutional Neural Networks (CNNs) are better suited to derive boundary likelihoods for visible object outlines, and introduces additional functionalities for the interactive delineation.

Finally, for step (iii), we introduce more intuitive and comprehensive delineation functionalities. While we tested the previous workflow on two UAV ortho-images in Germany, and delineated road outlines, we now test our workflow on imagery covering much larger extents and compare the results to cadastral boundaries. In this study, we test our workflow on UAV and aerial imagery of 0.02–0.25 m resolution from Kenya, Rwanda and Ethiopia covering 722 visible parcels.

Our new functionalities for the interactive delineation address cases for which the boundary classification fails, or is not necessary. Boundary classification comes into play in cases of over-segmentation, when many object outlines exist. Then, the delineator has to choose which lines demarcate the cadastral boundary. Support comes from the lines' boundary likelihood predicted by RF or CNN. In this study, we introduce functionalities that allow connecting image segmentation lines to cadastral boundaries, regardless of their boundary likelihood.

## 2. Materials and Methods

### 2.1. Image Data

An aerial image of 0.25 m Ground Sample Distance (GSD) of a rural scene in Ethiopia is used (Figure 2a,b). The local agricultural practice consists mostly of smallholder farming. The image was captured during the dry season around March. The crops within our study area consist mostly of millet, corn, and a local grain called teff. Since the crops are in the beginning of an agricultural cycle, they do not cover the visible cover parcel boundaries. The cadastral reference data cover 33 km$^2$ containing 9,454 plots with a median size of 2500 m$^2$. The cadastral reference data is derived through on-screen manual delineation from the aerial image. In case of uncertainty or invisible boundaries, the boundary is captured together with land owners in the field using high-resolution total stations. For a later assessment, in which we compare our approach to on-screen manual delineation, additional Unmanned Aerial Vehicle (UAV) data from Kenya and Rwanda is used (Figure 2c,d). The UAV data in Rwanda have a GSD of 0.02 m, and were captured with a rotary-wing Inspire 2 (SZ DJI Technology Co., Ltd., Shenzhen, Guangdong, People's Republic of China). The UAV data in Kenya have a GSD of 0.06 and were captured with a fixed-wing DT18 (Delair-Tech, Delair, Toulouse, France).

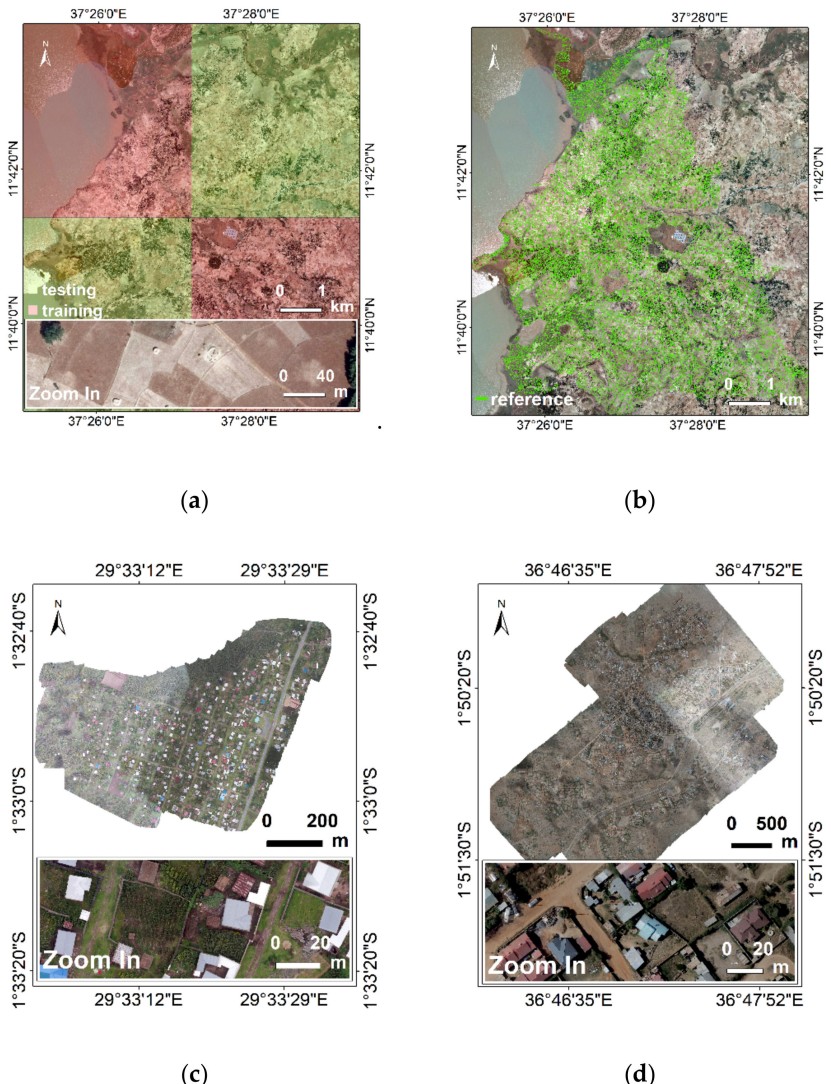

**Figure 2.** (**a**) Aerial image of 0.25 m Ground Sample Distance (GSD) for a rural scene in Ethiopia, divided into areas for training and testing our approach before comparing results to (**b**) the cadastral reference. Unmanned Aerial Vehicle (UAV) images for peri-urban scenes in (**c**) Rwanda (0.02 m GSD), and (**d**) Kenya (0.06 m GSD) to compare automated to manual delineation.

## 2.2. Boundary Mapping Approach

The boundary mapping approach refers to the one we previously described [22]. In the following, modifications and the data-dependent implementation of the three workflow steps are described. The source code is publically available [23].

Image segmentation is based on Multiresolution Combinatorial Grouping (MCG) [24], which delivers closed contours, capturing the outlines of visible objects. To run the original MCG implementation, the Ethiopian aerial image is tiled to 20 tiles of 8000 × 8000 pixels. The parameter k regulating over- and under-segmentation is set to produce over-segmentation (k = 0.1). This setting creates outlines around the majority of visible objects. Tests with parameters (k = 0.3 and k = 0.5) resulting in less over-segmentation show that visible object outlines are partly missed, while irrelevant lines around small objects are still produced. To reduce the number of irrelevant lines produced through over-segmentation, the lines are simplified through filtering (Figure 3): Lines around areas smaller than 30 m$^2$ are merged to the neighboring segments, which reduces the line count by 80% to 600,000 lines.

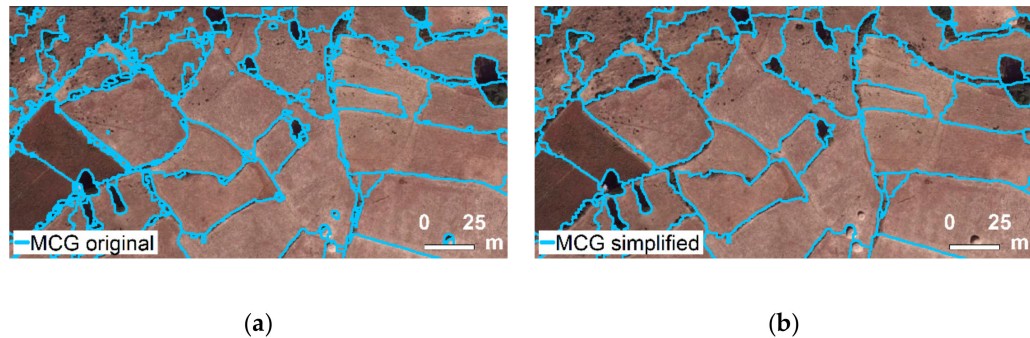

(**a**)                                                                (**b**)

**Figure 3.** Multiresolution Combinatorial Grouping (MCG) image segmentation lines around visible objects (**a**) before and (**b**) after simplification reducing the line count by 80%.

According to our visual inspections, this post-processing removes artefacts in the segmentation results and keeps outlines of large objects being more relevant for cadastral mapping. For the high-resolution data from Rwanda and Kenya, we proceed similarly by tiling the data and setting k = 0.4 and k = 0.3, respectively.

Boundary classification is applied to the post-processed 600,000 MCG lines. We investigate two machine learning approaches to derive the boundary likelihood per MCG line: Random Forest (RF) and Convolutional Neural Networks (CNN). Both require the labeling of training data as 'boundary' and 'not boundary'. The training data for RF consist of lines, that for CNN of image tiles. For both approaches, the cadastral reference is buffered by a radius of 0.4 m. This size accounts for inaccuracies in the cadastral reference and the ortho-image, enlarges the number of 'boundary' samples, and is identical to the one applied to derive hand-crafted RF features. For both approaches, the ratio between training and testing data is set to 50%. The number of 'boundary' and 'not boundary' training samples is balanced to 1:1 by randomly under-sampling 'not boundary' tiles (Table 1). The areas for training and testing are randomly selected and large to minimize the number of lines at the borders of each area that are clipped and of limited use for further analysis (Figure 2). The boundary likelihood predicted by both approaches represents the probability ($\hat{y}$) of a line being 'boundary':

$$boundary\ likelihood\ [0;1] = \hat{y}_{boundary} \tag{1}$$

**Table 1.** Distribution of training and testing data for boundary classification based on Random Forest (RF) and Convolutional Neural Networks (CNN).

|  | RF Classification | | CNN Classification | |
|---|---|---|---|---|
|  | Number of Lines | | Number of Tiles | |
| Label | Training | Testing | Training | Testing |
| 'boundary' | 12,280 (50%) | 9,742 (3%) | 35,643 (50%) | 34,721 (4%) |
| 'not boundary' | 12,280 (50%) | 280,108 (97%) | 34,665 (50%) | 746,349 (96%) |
| Σ | 24,560 | 289,850 | 70,308 | 781,070 |

RF classification is applied as shown in Figure 4 [22]. Instead of manually labeling lines for training, a line is now automatically labeled as 'boundary' when it overlaps with the cadastral reference buffer of 0.4 m by more than 50%. This value aligns with the threshold at which a CNN-derived result is labeled as 'boundary' or 'not boundary'. Since no DSM information is available for the study area, the feature dsm grad is not calculated.

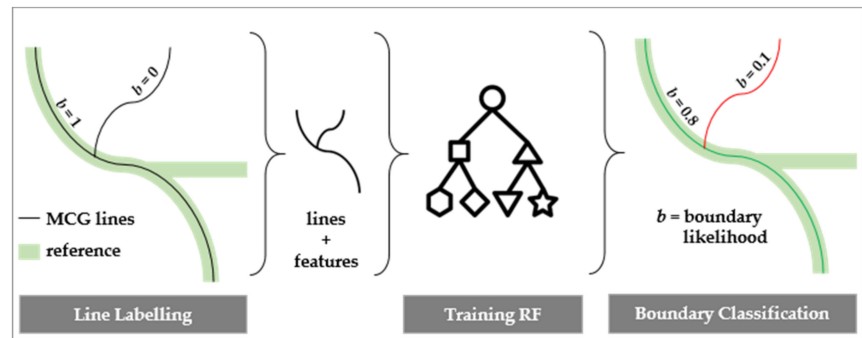

**Figure 4.** Boundary line classification based on Random Forest (RF) to derive boundary likelihoods for MCG lines.

CNN classification is investigated by training state-of-the-art tile-based CNNs (Figure 5). We reformulate our problem of generating boundary likelihoods for MCG lines to be solvable by a tile-based CNN as follows: At first, image tiles of $224 \times 224$ pixels centered on an MCG line are cropped from the ortho-image. $224 \times 224 \times 3$ is the standard size of images required by the used CNN. A tile is labeled as 'boundary' if the center pixel covering an MCG line overlaps with the cadastral reference buffer. A tile is created every 5 m along an MCG line. Decreasing this distance would increase the overlap, and thus the redundancy, of the image content per tile. Increasing this distance would reduce the number of tiles and thus the number of training data. With these settings, we generate 1.5 million tiles surrounding MCG pixels of which 5% are labeled as 'boundary' and 95% as 'not boundary'. After training, the CNN predicts boundary likelihoods for unseen testing areas (Figure 2a). The likelihoods of all tiles per MCG line are averaged based on the 97th percentile. This value aligns with the distribution of 'boundary' and 'not boundary' lines in the training data (Table 1). We use a pre-trained CNN architecture. We apply transfer learning by adding additional trainable layers: A global spatial average pooling layer, a fully connected layer with rectified linear unit (ReLU) activation, a dropout layer and a logistic layer with softmax activation. Only these last layers are trainable. We investigate using different pre-trained CNN architectures, including the Visual Geometry Group (VGG) [25], ResNet [26], Inception [27], Xception [28], MobileNet [29] and DenseNet [30], as well as different hyper-parameter settings on the learning optimizer, the depth of the fully connected layer and the dropout rate.

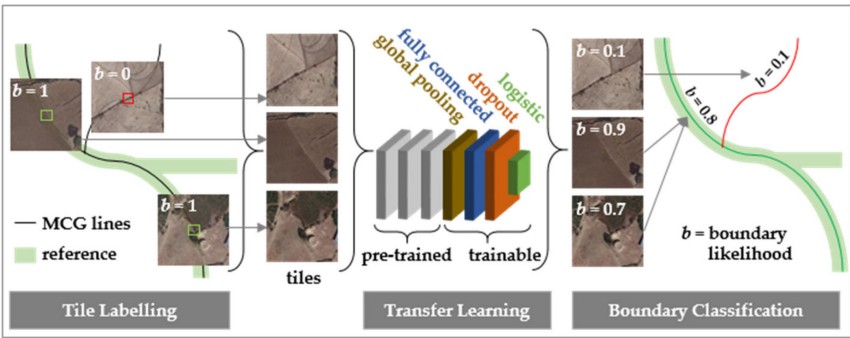

**Figure 5.** Boundary line classification based on Convolutional Neural Networks (CNNs) to derive boundary likelihoods for MCG lines.

Interactive delineation supports the creation of final cadastral boundaries. In comparison to our previous study [22], we now include more functionalities to delineate parcels (Table 2) and redesigned the Graphical User Interface (GUI). The interactive delineation is implemented in the open source geographic information system QGIS [31] as BoundaryDelineation plugin [32]:

**Table 2.** Delineation functionalities of BoundaryDelineation QGIS plugin.

| Functionality | Description |
| --- | --- |
| Connect around selection | Connect lines surrounding a click or selection of lines (Figure 6a,b) |
| Connect lines' endpoints | Connect endpoints of selected lines to a polygon, regardless of MCG lines (Figure 6c) |
| Connect along optimal path | Connect vertices along least-cost-path based on a selected attribute, e.g., boundary likelihood (Figure 6d) |
| Connect manual clicks | Manual delineation with the option to snap to input lines and vertices |
| Update edits | Update input lines based on manual edits |
| Polygonize results | Convert created boundary lines to polygons |

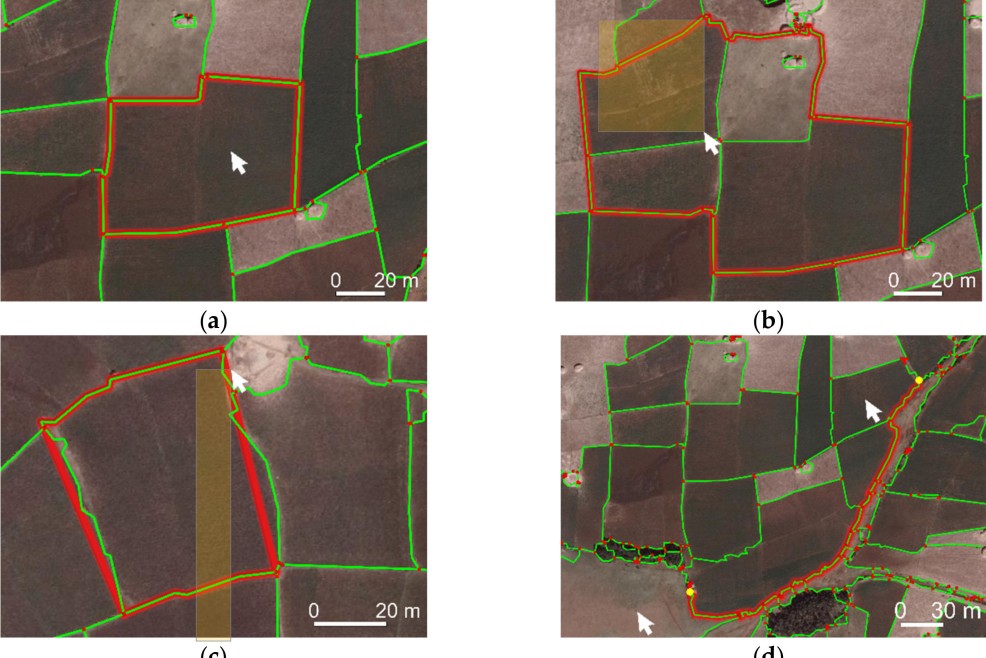

(**a**)　　　　　　　　　　　　(**b**)

(**c**)　　　　　　　　　　　　(**d**)

**Figure 6.** Interactive delineation functionalities: (**a**) Connect lines surrounding a click, or (**b**) a selection of lines. (**c**) Close endpoints of selected lines to a polygon. (**d**) Connect lines along least-cost-path.

*2.3. Accuracy Assessment*

The accuracy assessment investigates multiple aspects of our workflow, each requiring a different analysis:

CNN Architecture: This analysis aims to optimize the CNN architecture by considering loss and accuracy for training and validation data per epoch. The curves for training loss and validation loss, as well as for training accuracy and validation accuracy, are expected to converge with incremental epochs. Loss is the summation of errors made for each example in training, and should be minimized. We use cross-entropy loss that increases as the predicted probability ($\hat{y}_i$) diverges from the actual label ($y_i$):

$$cross-entropy\ loss = -(y_i \log(\hat{y}_i) + (1 - y_i) \log(1 - \hat{y}_i)) \tag{2}$$

All predictions < 0.5 are considered as 'not boundary', those ≥ 0.5 as 'boundary'. This results in a confusion matrix showing the number of tiles being False Positive (FP), True Positive (TP), False Negative (FN) and True Negative (TN). From this matrix, the accuracy is derived as the sum of correctly classified tiles divided by all tiles:

$$accuracy\ [0;1] = \frac{TP + TN}{TP + FP + FN + TN} \tag{3}$$

RF vs. CNN Classification: This analysis compares the boundary likelihood obtained through RF and CNN to the percentage to which an MCG line overlaps with the cadastral reference. Both are buffered with a radius of 0.4 m. The area of their overlap in relation to the entire MCG buffer area represents the percentage of overlap:

$$overlap~[0;1] ~=~ \frac{area_{MCG-buffer} ~\cap~ area_{cadastral-buffer}}{area_{MCG-buffer}} \tag{4}$$

We investigate whether lines that should get a boundary likelihood > 0, i.e., those that fall within the cadastral reference buffer, are assigned a boundary likelihood > 0:

$$recall~[0;1] = \frac{TP}{TP + FN} \tag{5}$$

Then, we check whether the assigned boundary likelihood is valid, i.e., whether it is equal to the line's overlap with the cadastral reference buffer. This is indicated by the precision that captures the ratio of lines having a boundary likelihood that aligns with overlap to the sum of lines having a correct or too positive boundary likelihood:

$$precision~[0;1] = \frac{TP}{TP + FP} \tag{6}$$

Since the boundary likelihood captures the probability of a line being a 'boundary' line, a high boundary likelihood should go along with a high overlap between the MCG and cadastral reference buffer:

$$overlap~[0;1]~boundary~likelihood~[0;1] \tag{7}$$

Both values are not expected to be identical, which can be influenced by altering the buffer size. Our focus is on comparing RF to CNN, and secondarily on the boundary likelihood itself. Results are considered only in areas for testing in which we have cadastral reference data (Figure 2).

Manual vs. Automated Delineation: This analysis compares the time and number of clicks required to delineate visible boundaries, once manually, and once with the automated approach. Manual delineation refers to delineating parcels based on the ortho-image without further guidance. Automated delineation refers to our approach, including RF or CNN classification depending on which approach shows superior results in this study. All delineations should fall within the cadastral reference buffer of the 0.4 m radius. The buffer size represents the local accepted accuracy for cadastral delineation and falls within the 2.4 m proposed for rural areas by the International Association of Assessing Officers (IAAO) [33].

The comparison is conducted for a rural area in Ethiopia and two peri-urban areas in Rwanda and Kenya (Figure 2). No urban area is selected, as indirect surveying relies on the existence of visible boundaries, which are rare in densely populated areas. Furthermore, indirect surveying in urban areas saves less logistics for field surveys, due to smaller parcel sizes. Only parcels for which all boundaries are visible, and thus detectable from the ortho-image, are kept for this analysis. Since no digital up-to-date cadastral reference exists for our areas in Kenya and Rwanda, cadastral reference data are created based on local knowledge in alignment with visible boundaries.

## 3. Results

### 3.1. CNN Architecture

We first tested different pre-trained base CNNs (VGG, ResNet, Inception, Xception, MobileNet and DenseNet) to which we added trainable layers. The combined CNN model was trained with a batch size of 32 for 100 epochs. In the case of no learning, the training stopped earlier. Out of the 50% of balanced data used for training, we used 10% for validation. These data were not seen by the

network, but were used only to calculate loss and accuracy per epoch. These metrics and their curves looked most promising for VGG19 [25]. Then we applied the trained network to the remaining 50% of testing data. VGG19 is a 19 layer deep CNN developed by the Visual Geometry Group (VGG) from the University of Oxford (Oxford, Oxfordshire, UK). It is trained to classify images into 1000 object categories, such as keyboard, mouse pencil and many animals. It has learned high-level features for a wide range of images from ImageNet [34]. ImageNet is a dataset of over 15 million labeled high-resolution images with around 22,000 categories. Compared to other CNNs, VGG has shown to generalize well, compared to more complex and less deep CNN architectures [25].

We used VGG19 layers pre-trained for 20,024,384 parameters as a base model. Next, we modified hyper-parameters for VGG19 on the learning optimizer, the depth of the fully connected layer, and the dropout rate to optimize accuracy and loss. We used softmax as an activation function to retrieve predictions for tiles being 'not boundary' in the range [0, 1]. These values represent the weights for the later least-cost-path calculation. Sigmoid activation, which is a type of softmax for a binary classification problem, provided similar results in terms of accuracy and loss. However, it required more post-processing, as the resulting value in the range [0, 1] cannot be understood as described for softmax activation.

The aim was to maximize the accuracy for training and validation data, while minimizing loss. To avoid over-fitting, the curves for training and validation accuracy should not diverge, which was achieved by increasing the dropout rate from 0.5 to 0.8. To avoid under-fitting, the curve for training accuracy should not be below that of the validation accuracy, which was avoided by increasing the depth of the fully connected layer from 16 to 1,024. To avoid oscillations in loss, the learning rate was lowered from 0.01 to 0.001. Learning was stopped once the validation accuracy did not further improve. Results and observations derived from different hyper-parameter settings and different pre-trained base CNNs are provided in the Appendix A (Table A1).

We achieved the best results after training 8,242 parameters on four trainable layers added to 22 pre-trained VGG19 layers (Table 3). This led to a validation accuracy of 71% and a validation loss of 0.598 after 200 epochs (Figure 7). The accuracy could be increased by 1% after 300 epochs, with validation loss restarting to increase to 0.623. We conclude that optimal results are achieved after 200 epochs. 100 epochs halve the training time to 11 hours, whilst obtaining 1% less accuracy and a loss of 0.588. The implementation relies on the open source library Keras [35], and this is publically available [23]. All experiments are conducted on a machine having a NVIDIA GM200 (GeForce GTX TITAN X) GPU with 128 GB RAM (Nvidia Corporation, Santa Clara, CA, US).

**Table 3.** Settings for our fine-tuned CNN based on Visual Geometry Group 19 (VGG19).

| | Settings | Parameters |
|---|---|---|
| untrainable layers | VGG19 pre-trained on ImageNet | exclusion of final pooling and fully connected layer |
| trainable layers | pooling layer<br>fully connected layer<br>dropout layer<br>logistic layer | global average pooling 2D<br>Depth = 1024, activation = ReLu<br>dropout rate = 0.8<br>Activation = softmax |
| learning optimizer | stochastic gradient descent (SGD) optimizer | learning rate = 0.001<br>momentum = 0.9<br>decay = learning rate/epochs |
| training | shuffled training tiles and un-shuffled validation tiles | Epochs = max. 200<br>batch size = 32 |

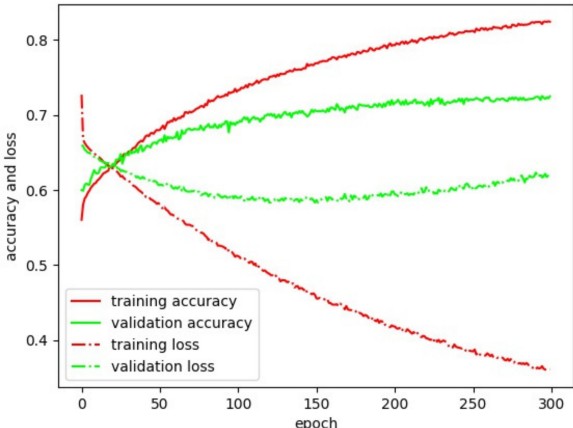

**Figure 7.** Accuracy and loss for our fine-tuned VGG19.

### 3.2. RF vs. CNN Classification

Of those lines that should get a boundary likelihood > 0, i.e., those that fall within the cadastral reference buffer, 100% for RF and 98% for CNN are assigned a boundary likelihood > 0 (Table 4). This means that both classifiers predict a boundary likelihood in the range [0, 1] when there is some overlap with the cadastral reference buffer.

**Table 4.** Is the boundary likelihood predicted for the correct lines?

| | | | Overlap | | | |
|---|---|---|---|---|---|---|
| | | | 0 | ]0; 1] | Σ | Σ % |
| **boundary likelihood** | RF | 0 | 535 | 265 | 800 | 0 |
| | | ]0; 1] | 150,583 | 59,123 | 209,706 | 100 |
| | | Σ | 151,118 | 59,388 | 210,506 | |
| | | Σ % | 72 | 28 | | 100 |
| | CNN | 0 | 7,560 | 1,794 | 9,354 | 4 |
| | | ]0; 1] | 145,558 | 57,594 | 201,152 | 96 |
| | | Σ | 151,118 | 59,388 | 210,506 | |
| | | Σ % | 72 | 28 | | 100 |

Next, we looked at how valid the boundary likelihood is, i.e., whether its value is equal to the line's overlap with the cadastral reference buffer. For this we excluded lines having no overlap with the cadastral reference buffer, i.e., those having an overlap = 0. We grouped the remaining lines to compare boundary likelihood and overlap values (Table 5). For RF-derived boundary likelihoods, we obtained an accuracy of 41% and a precision of 49%. For CNN-derived boundary likelihoods, we obtained an accuracy of 52% and a precision of 76%. The percentage of lines per value interval of 0.25 for the same boundary likelihood and overlap value deviated on average by 15% for RF and by 7% for CNN (Table 5).

Overall, CNN-derived boundary likelihoods obtained a similar recall, a higher accuracy, and a higher precision (Table 4). The percentage of lines for different ranges of boundary likelihoods represented the distribution of overlap values more accurately (Table 5). Even though the values of overlap and boundary likelihood do not express the same, they provide a valid comparison between RF- and CNN-derived boundary likelihoods. We consider CNN-derived boundary likelihoods a better input for the interactive delineation, and continue the accuracy assessment for a boundary classification based on CNN.

**Table 5.** How correct is the predicted boundary likelihood?

|  |  |  | overlap |  |  |  |  |  |
|---|---|---|---|---|---|---|---|---|
|  |  | 0 | ]0; 0.25] | ]0.25; 0.5] | ]0.5; 0.75] | ]0.75; 1] | ∑ | ∑ % |
| boundary likelihood | RF | ]0; 0.25] |  | 15,176 | 3,633 | 481 | 95 | 19,385 | 32 |
|  |  | ]0.25; 0.5] | 151,118 | 11,553 | 5633 | 2178 | 730 | 20,094 | 34 |
|  |  | ]0.5; 0.75] |  | 6827 | 4849 | 3120 | 1617 | 16,413 | 28 |
|  |  | ]0.75; 1] |  | 973 | 1002 | 813 | 708 | 3496 | 6 |
|  |  | ∑ |  | 34,529 | 15,117 | 6592 | 3150 | 59,388 |  |
|  |  | ∑ % | / | 58 | 26 | 11 | 5 |  | 100 |
|  | CNN | ]0; 0.25] |  | 26,546 | 10,472 | 4305 | 1981 | 43,304 | 73 |
|  |  | ]0.25; 0.5] | 151,118 | 5974 | 3307 | 1534 | 765 | 11,580 | 19 |
|  |  | ]0.5; 0.75] |  | 1751 | 1177 | 655 | 328 | 3,911 | 7 |
|  |  | ]0.75; 1] |  | 258 | 161 | 97 | 77 | 593 | 1 |
|  |  | ∑ |  | 34,529 | 15,117 | 6591 | 3151 | 59,388 |  |
|  |  | ∑ % | / | 58 | 26 | 11 | 5 |  | 100 |

### 3.3. Manual vs. Automated Delineation

Indirect surveying, comprising of manual or automated delineation, both rely on visible boundaries. Before comparing manual to automated delineation, we filtered the cadastral reference data for Ethiopia (Figure 2b) to contain visible parcels only. We kept only those parcels for which all boundary parts were visually demarcated. As in Kohli et al. [4], we consider only fully closed polygons that are entirely visible in the image. From the original cadastral reference data, we kept 38% of all parcels for which all boundaries were visible. In Kohli et al. [4], the portion of fully visible parcels has been reported to average around 71% of all cadastral parcels in rural Ethiopian areas. We can confirm 71% for parts of our study area that cover smallholder farms. Cadastral data for Rwanda and Kenya were delineated based on local knowledge in alignment with visible boundaries. As for Ethiopia, only fully closed and visible parcels were considered. The mean size of our visible parcels amounts to 2,725 m$^2$ for Ethiopia, 656 m$^2$ for Rwanda, and 730 m$^2$ for Kenya.

When manually delineating visible boundaries, we observed how tiring a task this manual delineation is: The delineator has to continuously scan the image for visible boundaries to then click precisely and repeatedly along the boundary to be delineated. Apart from the visual observation of the ortho-image, the delineator has no further guidance on where to click. Each parcel is delineated the same way, which makes it a highly repetitive task that exhausts the eyes and fingers in no time.

When comparing manual to automated delineation, this impression changes: The delineator now has lines and vertices to choose from, which can be connected automatically using multiple functionalities (Table 2, Figure 6). Complex, as well as simple, parcels require fewer clicking when delineating with the automated approach: To follow a curved outline, manual delineation requires frequent and accurate clicking while zooming in and out. Automated delineation requires clicking on vertices covering the start and endpoint once, before they are automatically connected precisely following object outlines (Figure 6d). Similarly, the automated delineation is superior for simple rectangular parcels: While manual delineation requires accurate clicking on each of the at least four corners of a rectangle, automated delineation allows clicking once somewhere inside the rectangle to retrieve its outline (Figure 8a).

However, choosing the optimal functionality can be time-consuming, especially in cases of fragmented MCG lines obtained from high-resolution UAV data. We assume that the time for automated delineation can be reduced through increased familiarity with all functionalities and by further developing their usability, e.g., by keyboard shortcuts.

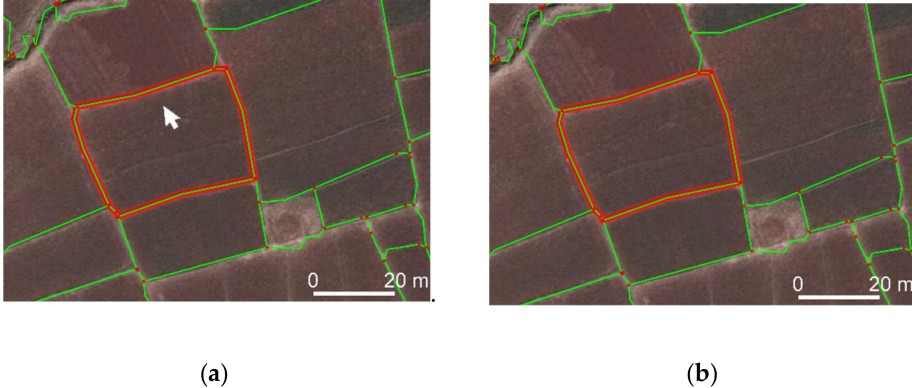

<div align="center">(<b>a</b>)           (<b>b</b>)</div>

**Figure 8.** (**a**) Automated delineation requires clicking once somewhere in the parcel, while manual delineation requires precise clicking at least four times on each corner. (**b**) Boundaries partly covered or delineated by vegetation impede indirect surveying and limit the effectiveness of our automated delineation compared to manual delineation.

Automated delineation required fewer clicks for our rural and peri-urban study areas (Table 6). Only those parcels for which one of our functionalities was more effective than manual delineation are considered for the automated delineation, amounting to 40–58% of all visible parcels. The effectiveness of manual delineation is considered for all 100% of the visible parcels. By maximizing the number of delineated parcels, we aimed to minimize the effect of unusual parcels that required much effort to delineate manually. We expect the measures that we obtained for the manual delineation to be similar for the 40–58% of parcels considered for the automated delineation. For the remaining parcels, MCG lines were either not available, or not aligning enough with the reference data. Manually delineating these parcels with the plugin requires the same number of clicks and time as conventional manual delineation, but is partly less tiring, as the delineation can be snapped to the MCG lines and vertices.

**Table 6.** Does automated delineation cost less effort?

| | **Manual Delineation** | | | **Automated Dlineation** | | |
|---|---|---|---|---|---|---|
| | Parcel Count | $\frac{time}{parcel}$ [s] | $\frac{clicks}{parcel}$ | Parcel Count | $\frac{time}{parcel}$ [s] | $\frac{clicks}{parcel}$ |
| Ethiopia (rural) | 350 | 13 | 10 | 181 (52%) | 8 | 2 |
| Rwanda (peri-urban) | 100 | 12 | 7 | 40 (40%) | 25 | 5 |
| Kenya (peri-urban) | 272 | 11 | 5 | 157 (58%) | 10 | 4 |

Nevertheless, the lines and vertices can also impede the visibility: For our data from Rwanda and Kenya, the boundaries are not continuously visible. The partly vegetation-covered boundaries result in zigzagged and fragmented MCG lines (Figure 8b). Additionally, visible boundaries with low contrast were partly missed by MCG image segmentation. In both cases, the advantages of automated delineation are limited.

We claimed that the least-cost-path based on the boundary likelihood is beneficial to delineate long and curved outlines [21,22]. For the Ethiopian data, we now barely made use of the boundary likelihood: For the often small and rectangular parcels, connecting all lines surrounding a click or a selection of lines was more efficient. For areas with few fragmented, long or curved outlines, the workflow is assumed to be of similar effectiveness when leaving out the boundary classification. To include the boundary classification is beneficial when boundaries are demarcated, e.g., by long and curved boundaries, such as roads, waterbodies, or vegetation.

For our data from Kenya and Rwanda, we omitted the boundary classification, since we hardly used it for the Ethiopian data. The least-cost-path, for which a weight attribute can be selected in the plugin interface, used line length instead of boundary likelihood. Since the boundaries differ from

the boundaries in the Ethiopian scene, the CNN would need to be retrained or fine-tuned for the new boundary types. Retrieving CNN-derived boundary likelihoods for these UAV data would require further experiments on whether and how to rescale tiles to 224 × 224 pixels, while providing context comparable to our aerial tiles (Figure 5).

Overall, the automated delineation provided diverse functionalities for different boundary types (Table 7), which made delineation less tiring and more effective (Table 6). Improvements to manual delineation were the strongest for parcels fully surrounded by MCG lines. Such parcels were mostly found in the Ethiopian rural scene, where boundaries aligned with agricultural fields. In the Rwandan scene, automated delineation was time-consuming, since the boundaries were not demarcated consistently. Selecting and joining fragmented MCG lines required more careful visual inspection compared to the rural Ethiopian scene. In the Kenyan scene, the boundaries were less often covered by vegetation, and thus were in general better visible. Compared to the rural Ethiopian scene, the automated delineation still required more zooming, as boundaries were demarcated by more diverse objects.

**Table 7.** Which plugin functionality to use for which boundary type?

| Functionality | Boundary Type | Boundary $\hat{=}$ Segmentation | Example Boundary |
|---|---|---|---|
| Connect around selection | complex or rectangular | yes | agricultural field |
| Connect lines' endpoints | small or rectangular | partly | vegetation-covered |
| Connect along optimal path | long or curved | yes | curved river |
| Connect manual clicks | fragmented or partly invisible | no or partly | low-contrast |

## 4. Discussion

### 4.1. Working Hypothesis: Improving Boundary Mapping Approach

Compared to our previous workflow [21], we improved each of the three workflow steps. For image segmentation, we remove the previous need to reduce the image resolution for images larger than 1000 × 1000 pixels, and we introduce a filtering step that allows us to limit over-segmentation by reducing the number of segment lines by 80%. For boundary classification, we implement Convolutional Neural Networks (CNNs) and thereby eliminate the previous need for Random Forest (RF) hand-crafted feature generation. For interactive delineation, we develop two additional delineation functionalities ('Connect around selection', 'Connect lines' endpoints'), we develop an attribute selection for the least-cost-path functionality ('Connect along optimal path') and redesign the GUI to be more intuitive. While we previously tested our approach on road outlines only, we now show advantages compared to manual delineation for cadastral mapping, which includes various object types. The number of clicks per 100 m compared to manual delineation was previously reduced by 76% and 86%, respectively, when delineating roads from two UAV images. Now we applied our approach to delineate 378 visible cadastral boundaries from UAV and aerial imagery of larger extents, while requiring on average 80% fewer clicks compared to manual delineation.

### 4.2. Working Hypothesis: CNN vs. RF

By reformulating our problem to be solvable by a CNN, we have investigated integrating a more state-of-the-art approach in our previously proposed boundary delineation workflow [21]. A deep learning CNN was assumed to be superior to a machine learning RF, as CNNs require no hand-crafted features, and can be trained incrementally. This starting hypothesis holds true: Even though pre-trained on images from computer vision, transfer-learning a CNN on remote sensing data provided more accurate predictions for boundary likelihoods compared to RF. Our successful integration reduces the effect of possibly meaningless or biased hand-crafted features, and increases the degree of automation

of our approach. However, when conducting the final workflow step, i.e., interactive delineation, we found that we seldom made use of the boundary likelihood. We reduced over-segmentation, due to post-processing the image segmentation. This, in combination with new interactive delineation functionalities, is more effective than manual delineation for regular-shaped parcels surrounded by visible boundaries. The delineation functionality that uses boundary likelihood is beneficial for long or curved boundaries, which was rare in our study areas.

### 4.3. Limitations & Future Work

When training a network to predict boundary likelihoods for visible object outlines, our training data based on cadastral reference are beneficial, as it is available without further processing. The data have little bias, as no human annotator with domain knowledge is required [36]. However, the data could be improved: Cadastral data contain invisible boundaries not detectable by MCG. To limit training data to visible boundaries would match better with what the network is expected to learn, and thus increase achievable accuracy metrics. When deciding whether to use RF or CNN for boundary classification, one needs to balance feature extraction for RF [37] against training data preparation and computational requirements for CNN [18]. In cases of limited training data for CNN, our CNN-based boundary classification may be adopted by data augmentation and re-balancing class weights. One advantage of our RF-based boundary classification is that it contains a feature capturing 3D information from a Digital Surface Model (DSM) [21]. 3D information still needs to be included in the CNN-based boundary classification. Compared to computer vision, the amount and size of benchmark image data are marginal: Existing benchmarks cover aerial data for urban object classification [38] and building extraction [39], satellite imagery for road extraction, building extraction and land cover classification [40], as well as satellite and aerial imagery for road extraction [41]. Such benchmarks in combination with open data initiatives for governmental cadastral data [42], aerial imagery [43] and crowdsourced labeling [44–46] may propel deep learning frameworks for cadastral boundary delineation, i.e., cadastral intelligence. Instead of using a VGG pre-trained on ImageNet, our approach could then be trained on diverse remote sensing and cadastral data, resulting in a possibly more effective and scalable network.

Despite the shown advances, automating cadastral boundary delineation is not at its end. Identifying areas in which a large portion of cadastral boundaries is visible, and for which high-resolution remote sensing and up-to-date cadastral data are available in digital form, still impedes methodological development. Future work could investigate the approach's applicability for invisible boundaries, that are marked before UAV data capture, e.g., with paint or other temporary boundary markers. In this context, the degree to which the approach can support participatory mapping could also be investigated. Furthermore, research needs to be done on how to align innovative approaches with existing technical, social, legal and institutional frameworks in land administration [47–49]. We are currently pursuing this by developing documentation and testing material [50] that enables surveyors and policy makers in land administration to easily understand, test and adapt our approach.

### 4.4. Comparison to Previous Studies

How we reformulated our problem to be solvable by a tile-based CNN has been similarly proposed in biomedical optics [51]. Fang et al. crop tiles centered on retinal boundary pixels and train a CNN to predict nine different boundary labels. Correspondingly labeled pixels are connected with a graph-based approach. To transfer the latter to our case, we may investigate whether connecting tiles of similar boundary likelihood can omit the need for an initial MCG image segmentation: By using Fully Convolutional Networks (FCNs) [52] each pixel of the input image would be assigned a boundary likelihood, which can be connected using Ultrametric Contour Maps (UCMs) [53] included in MCG [54]. Connecting pixels of corresponding boundary likelihoods could also be realized by using MCG-based contour closure [55], line integral convolution [56], or template matching [57].

Alternatively, the topology of MCG lines can be used to sort out false boundary likelihoods before aggregating them per line: This could be realized by not shuffling training data, and thus maintaining more context information per batch, or by using graph-based approaches such as active contour models [58] suggested for road detection [59,60], or region-growing models suggested for RF-based identification of linear vegetation [61].

Predicting the optimal MCG parameter k per image may also be achieved with CNNs. Depending on whether an area is, e.g., rural or urban, cadastral parcels vary in size and shape. Larger parcels demand less over-segmentation and a higher k. Similarly, our high-resolution UAV data required a higher k, i.e., 0.3 and 0.4 as compared to 0.1 for the aerial data. Challenges to be addressed are training with data from multiple sensors, varying parcel sizes in training and automatically labeling data with the optimal segmentation parameter k.

## 5. Conclusions

We have introduced a workflow that simplifies the image-based delineation of visible boundaries to support the automated mapping of land tenure from various sources' remote sensing imagery. In comparison to our previous work [21], the approach is now more automated and more accurate due to the integration of CNN deep learning, compared to RF machine learning. For RF-derived boundary likelihoods, we obtained an accuracy of 41% and a precision of 49%. For CNN-derived boundary likelihoods, we obtained an accuracy of 52% and a precision of 76%. CNNs eliminate the need to generate hand-crafted features required for RF. Furthermore, our approach has proven to be less tiring and more effective compared to manual delineation, due to the decreased over-segmentation and our new delineation functionalities. We limit over-segmentation by reducing the number of segment lines by 80% through filtering. Through the new delineation functionalities, the delineation effort per parcel requires 38% less time and 80% fewer clicks compared to manual delineation. The approach works on data from different sensors (aerial and UAV) of different resolutions (0.02–0.25 m). Advantages are strongest when delineating in rural areas due to the continuous visibility of monotonic boundaries. Manual delineation remains superior in cases where the boundary is not fully visible, i.e., covered by shadow or vegetation. While our approach has been developed for cadastral mapping, it can also be used to delineate objects in other application fields, such as land use mapping, agricultural monitoring, topographical mapping, road tracking, or building extraction.

**Author Contributions:** Conceptualization, S.C., M.K., M.Y.Y. and G.V.; methodology, S.C., M.K., M.Y.Y. and G.V.; software, S.C.; validation, S.C., M.K., M.Y.Y. and G.V.; formal analysis, S.C.; investigation, S.C.; resources, S.C.; data curation, S.C.; writing—original draft preparation, S.C.; writing—review and editing, S.C., M.K., M.Y.Y. and G.V.; visualization, S.C.; supervision, M.K., M.Y.Y. and G.V.; project administration, M.K.; funding acquisition, G.V.

**Funding:** This research was funded by Horizon 2020 program of the European Union (project number 687828).

**Acknowledgments:** We are grateful for the support of our African project partners INES Ruhengeri (Rwanda), Bahir Dar University (Ethiopia), Technical University of Kenya, and Esri Rwanda to support the data capture, which was guided and further processed by Claudia Stöcker from University of Twente. Berhanu Kefale Alemie from Bahir Dar University provided aerial data and reference data, as well as knowledge about local land administration situation.

**Conflicts of Interest:** The authors declare no conflict of interest.

## Appendix A

**Table A1.** Results obtained on validation data for different fine-tuned CNNs. The one used for further analysis in our study is outlined in green. The legend text corresponds to that of Figure 7.

| Parameter | Value | Acc. | Loss | Plot |
|---|---|---|---|---|
| base model | VGG19 | 0.607 | 0.654 |  |
| dense layer depth | / | | | |
| dense layer depth | 16 | | | |
| dropout rate | 0.5 | | | |
| learning rate | 0.01 | | | |
| base model | VGG19 | 0.705 | 0.66 |  |
| dense layer depth | / | | | |
| dense layer depth | 1024 | | | |
| dropout rate | 0.5 | | | |
| learning rate | 0.01 | | | |
| base model | VGG19 | 0.693 | 1.632 |  |
| dense layer depth | 512 | | | |
| dense layer depth | 16 | | | |
| dropout rate | 0.5 | | | |
| learning rate | 0.01 | | | |
| base model | VGG19 | 0.613 | 0.643 |  |
| dense layer depth | / | | | |
| dense layer depth | 16 | | | |
| dropout rate | 0.5 | | | |
| learning rate | 0.001 | | | |
| base model | VGG19 | 0.615 | 0.646 |  |
| dense layer depth | / | | | |
| dense layer depth | 16 | | | |
| dropout rate | 0.2 | | | |
| learning rate | 0.01 | | | |
| base model | VGG19 | 0.6 | 0.656 |  |
| dense layer depth | 16 | | | |
| dense layer depth | 16 | | | |
| dropout rate | 0.8 | | | |
| learning rate | 0.001 | | | |
| base model | VGG16 | 0.667 | 0.608 |  |
| dense layer depth | / | | | |
| dense layer depth | 1024 | | | |
| dropout rate | 0.8 | | | |
| learning rate | 0.001 | | | |

**Table A1.** *Cont.*

| Parameter | Value | Acc. | Loss | Plot |
|---|---|---|---|---|
| base model | VGG19 | | | |
| dense layer depth | / | | | |
| dense layer depth | 1024 | 0.692 | 0.586 | |
| dropout rate | 0.8 | | | |
| learning rate | 0.001 | | | |
| base model | VGG19 | | | |
| dense layer depth | / | | | |
| dense layer depth | 1024 | 0.733 | 1.205 | |
| dropout rate | 0.8 | | | |
| learning rate | 0.001 | | | |
| base model | ResNet50 | | | |
| dense layer depth | / | | | |
| dense layer depth | 16 | 0.571 | 0.742 | |
| dropout rate | 0.5 | | | |
| learning rate | 0.01 | | | |
| base model | ResNet50 | | | |
| dense layer depth | / | | | |
| dense layer depth | 1024 | 0.561 | 2.367 | |
| dropout rate | 0.5 | | | |
| learning rate | 0.01 | | | |
| base model | ResNet50 | | | |
| dense layer depth | 512 | | | |
| dense layer depth | 16 | 0.546 | 3.86 | |
| dropout rate | 0.5 | | | |
| learning rate | 0.01 | | | |
| base model | ResNet50 | | | |
| dense layer depth | / | | | |
| dense layer depth | 16 | 0.577 | 0.787 | |
| dropout rate | 0.5 | | | |
| learning rate | 0.001 | | | |
| base model | ResNet50 | | | |
| dense layer depth | / | | | |
| dense layer depth | 16 | 0.578 | 0.838 | |
| dropout rate | 0.2 | | | |
| learning rate | 0.01 | | | |
| base model | InceptionV3 | | | |
| dense layer depth | / | | | |
| dense layer depth | 1024 | 0.543 | 0.792 | |
| dropout rate | 0.8 | | | |
| learning rate | 0.001 | | | |

**Table A1.** *Cont.*

| Parameter | Value | Acc. | Loss | Plot |
|-----------|-------|------|------|------|
| base model | Xception | | | |
| dense layer depth | / | | | |
| dense layer depth | 1024 | 0.559 | 0.777 | |
| dropout rate | 0.8 | | | |
| learning rate | 0.001 | | | |
| base model | MobileNet | | | |
| dense layer depth | / | | | |
| dense layer depth | 1024 | 0.612 | 0.775 | |
| dropout rate | 0.8 | | | |
| learning rate | 0.001 | | | |
| base model | DenseNet201 | | | |
| dense layer depth | / | | | |
| dense layer depth | 1024 | 0.569 | 0.895 | |
| dropout rate | 0.8 | | | |
| learning rate | 0.001 | | | |

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
