# Peer review of "Application of Deep Learning for Delineation of Visible Cadastral Boundaries from Remote Sensing Imagery"

_remotesensing, doi:10.3390/rs11212505_

Round 1
Reviewer 1 Report
A very interesting article in which the authors improve the automation process in detecting cadastral boundaries. They examined the inclusion of the most modern approach (CNN - Convolutional Neural Networks) to the previously proposed scheme of the boundaries delineation. The authors' approach is now more automated due to the integration of the deep learning, which turns out to be less tiring and more effective in comparison with the manual sketching due to reduced segmentation and new sketching features. The research was conducted on both UAV data and aerial photographs of various resolutions. They are supported by an in-depth analysis of the obtained results, as well as the guidelines for further work on this issue.
The advantage of the method is that although the authors' approach was developed for cadastral mapping, it can also be used to delineate objects in other application fields, such as land use mapping, topographic mapping, road tracking or building extraction.
The publication contains editorial error that should be corrected before accepting it for printing:
in line 320- (Figure 6Error! Reference source not found.d)
The article meets the requirements of the „Remote Sensing” journal and can be accepted for publication after the suggested corrections have been included and does not require another re-review.
Author Response
Dear reviewer,
thanks for having taken the time to review our submission and for providing your feedback.
We have updated our manuscript and corrected the issue you pointed out in line 320.
Kind regards,
Sophie Crommelinck et al.
Reviewer 2 Report
This study describes significant steps forward in the (semi-)automation of land boundaries using aerial imagery and deep learning, with the intention of establishing a production-ready workflow. The technical background is well described and the results convincing, though the reader will probably have to go through the previous publications of the author cited in this study in order to fully understand the method and especially the advance beyond the state of the art. The text reads well and the English is flawless, although slightly less technical language might be justified since the study will be of interest to practitioners beyond remote sensing and machine learning science.
Some detailed comments on specific parts of the text are the following:
Line 2 Title: The title should be slightly modified in order to indicate that this is not the first study to attempt "Semi-Automated delineation of visible cadastral boundaries from remote sensing imagery" (which is in essence nearly the same as the title of Crommelinck 2018), eg to "Application of CNN deep learning to semi-automatic delineation of visible cadastral boudaries from airborne remote sensing imagery" or similar.
Line 10-20 Abstract: here you mention that you have previously introduced a method for this task. In addition to listing the accuracies obtained, I suggest you also mention already in the abstract the improvement in accuracy (and/or performance, user experience) compared to your previous method
Line 48-52: In addition to introducing the main workflow of the previously proposed system, please also include a statement about its accuracy here.
Line 56: Here I suggest you mention that in addition to these improvements, you evaluate the accuracy of the new algorithm and workflow on a different dataset of several hundred thousand line segments
Line 93: "Sometimes, the prediction is not correct or only one outline surrounds a parcel." please formulate more clearly what you mean by "only one outline surrounds a parcel". Do you mean only one outline segment is identified for the parcel and other parts of the boundary which should be there are missing?
Line 96 to 100: This is a rather generic formulation of your objective, which is well and good. However, if this study aims to be so comprehensive, please describe your previous studies that you cite and that are the basis of the current study in more detail (eg. Crommelinck, Höfle et al 2018, Crommelinck Koeva et al Arxiv etc). Alternatively, you can provide a more specific objective (that your goal is to advance this specific aspect of the general problem). I also recommend you further emphasize the objective by formatting it to be a separate subsection of section 1.
Line 103-104: please add some more details about the aerial image. Which season was it captured in and how does that compare to the local agricultural cycle? Which phase of cropping are the fields likely to be in, what are the main crops or management forms?
Line 109, page 3: referring to a submitted paper for technical description of capture and processing should be avoided. Please provide a few sentences describing the technical basics here, and should the other paper be accepted during the review process of this paper, you can still add the citation and shorten the text. Otherwise you risk publishing this paper earlier than the other one and the reader might not be able to find the necessary information because the paper you cite does not exist (yet). Please consult with the technical staff or production editor about the journal policy on (the formatting of) citation of material in review.
Line 116: typo: "Ethiopia is use" should be "Ethiopia is used."
Lines 237-238: "We used 10% of the balanced training data for validation" : this seems like a very small set of data used for validation. Please justify or explain.
Lines 270-272: It seems that you left some text from the Latex template file in the final manuscript text. I assume this was not on purpose - please remove.
Lines 300-308 "we filtered the cadastral reference dta for Ethiopia to contain visible parcels only etc": How much of the data was removed by this step? What do the 38% and the 71% refer to? Is this in pixels or parcels, or boundary segments, How much do you think it influences the final accuracy, including the less visible parcel boundaries? It would seem that the final accuracy you reach is strongly influenced by this step, which is not clearly documented. If you only keep the "easy cases" it would not be surprising that your accuracy is high. On the other hand, if you are comparing to hand digitized boundaries, they will also have errors where the parcel boundaries are not visible.
Line 320 "Figure 6Error! Reference source not found.d" : seem like this is a latex problem her
Line 376 Discussion: T discussion should quantitatively compare the accuracies and performance reached by the precursor of this method (described in Crommelinck, Höfle et al 2018) and the new method published here (even if the data used are not the same), adding in a short paragraph on this comparison. Also, the remaining sources of error and uncertainty should be discussed, ideally in a separate subsection. Please check the requirements for this section in the author guidelines.
In the discussion, you should discuss the implications of your findings not only in the context of technological development of boundary detection but also from the perspective of the user: will your advances really make a difference for the cause you state in the introduction ( recording land rights for security)? You might also want to mention the eventual relevance of this method for land parcel change detection in an EU agricultural subsidies monitoring context: in many countries the updating of the LPIS is still done by hand digitizing.
Line 421-456: Please see author guidelines here as well: "Conclusions should be self-contained, a reader should not have to read through the full paper to understand it". It would be essential that you quantitatively formulate the improvement in accuracy you reached compared to the previous findings.
Line 446-456: this section is more of a future outlook and should be a separate sub-section of the discussion instead of part of the conclusions.
Author Response
Dear reviewer,
thanks for having taken the time to review our submission and for providing your feedback.
We have updated our manuscript and addressed the issues you pointed as described per issue in orange below. Text marked in bold, is the text we added or modified.
Kind regards,
Sophie Crommelinck et al.
Line 2 Title: The title should be slightly modified in order to indicate that this is not the first study to attempt "Semi-Automated delineation of visible cadastral boundaries from remote sensing imagery" (which is in essence nearly the same as the title of Crommelinck 2018), eg to "Application of CNN deep learning to semi-automatic delineation of visible cadastral boudaries from airborne remote sensing imagery" or similar.
We modified the title to :
Application of CNN Deep Learning for Delineation of Visible Cadastral Boundaries from Remote Sensing Imagery
Line 10-20 Abstract: here you mention that you have previously introduced a method for this task. In addition to listing the accuracies obtained, I suggest you also mention already in the abstract the improvement in accuracy (and/or performance, user experience) compared to your previous method
We updated the abstract as follows:
We have previously introduced a boundary delineation workflow comprising image segmentation, boundary classification, and interactive delineation that we applied on UAV data to delineate roads. In this study, we improve each of these steps. For image segmentation, we remove the need to reduce the image resolution and we limit over-segmentation by reducing the number of segment lines by 80% through filtering. For boundary classification, we show how Convolutional Neural Networks (CNN) can be used for boundary line classification, thereby eliminate the previous need for Random Forest (RF) feature generation and achieve 71% accuracy. For interactive delineation, we develop additional and more intuitive delineation functionalities that cover more application cases. We test our approach on more varied and larger data sets by applying it to UAV and aerial imagery of 0.02 - 0.25 m resolution from Kenya, Rwanda and Ethiopia.
Line 48-52: In addition to introducing the main workflow of the previously proposed system, please also include a statement about its accuracy here.
We modified the text to:
For that workflow, the number of clicks per 100 m compared to manual delineation was reduced by up to 86%, while obtaining a similar localization quality
Line 56: Here I suggest you mention that in addition to these improvements, you evaluate the accuracy of the new algorithm and workflow on a different dataset of several hundred thousand line segments
We added:
While we tested the previous workflow on two UAV orthoimages in Germany and delineated road outlines, we now test our workflow on imagery covering much larger extents and compare the results to cadastral boundaries. In this study, we test our workflow on UAV and aerial imagery of 0.02 - 0.25 m resolution from Kenya, Rwanda and Ethiopia covering 722 visible parcels.
Line 93: "Sometimes, the prediction is not correct or only one outline surrounds a parcel." please formulate more clearly what you mean by "only one outline surrounds a parcel". Do you mean only one outline segment is identified for the parcel and other parts of the boundary which should be there are missing?
We deleted the sentence as we agree that it brings more confusion than clarification.
Line 96 to 100: This is a rather generic formulation of your objective, which is well and good. However, if this study aims to be so comprehensive, please describe your previous studies that you cite and that are the basis of the current study in more detail (eg. Crommelinck, Höfle et al 2018, Crommelinck Koeva et al Arxiv etc). Alternatively, you can provide a more specific objective (that your goal is to advance this specific aspect of the general problem). I also recommend you further emphasize the objective by formatting it to be a separate subsection of section 1.
We included a new subsection and made the comparison to previous work clearer:
1.2 Study Objective
The main goal of our research is to develop an approach that simplifies image-based cadastral mapping to support the automated mapping of land tenure. We pursue this goal by developing an automated cadastral boundary delineation approach applicable to remote sensing imagery. In this study, we describe our approach in detail, optimize its components, apply it to more varied and larger remote sensing imagery from Kenya, Rwanda and Ethiopia, test it’s applicability to cadastral mapping, and assess its effectiveness compared to manual delineation.
We previously proposed a semi-automated indirect surveying approach for cadastral mapping from remote sensing data. To delineate roads from UAV imagery with that workflow, the number of clicks per 100 m compared to manual delineation was reduced by up to 86%, while obtaining a similar localization quality [21]. The workflow consists of: (i) image segmentation to extract visible object outlines, (ii) boundary classification to predict boundary likelihoods for extracted segment lines, and (iii) interactive delineation to connect these lines based on the predicted boundary likelihood. In this study, we investigate improvements in all three steps (Figure 1). First, for step (i), we filter out small segments to reduce over-segmentation. Second, for step (ii), we replace hand-crafted features and line classification based on Random Forest (RF) by Convolutional Neural Networks (CNNs). Finally, for step (iii), we introduce more intuitive and comprehensive delineation functionalities. While we tested the previous workflow on two UAV orthoimages in Germany and delineated road outlines, we now test our workflow on imagery covering much larger extents and compare the results to cadastral boundaries. In this study, we test our workflow on UAV and aerial imagery of 0.02 - 0.25 m resolution from Kenya, Rwanda and Ethiopia covering 722 visible parcels.
Our new functionalities for the interactive delineation address cases for which the boundary classification fails or is not necessary. Boundary classification comes into play in cases of over-segmentation, when many object outlines exist. Then, the delineator has to choose which lines demarcate the cadastral boundary. Support comes from the lines’ boundary likelihood predicted by RF or CNN. In this study, we introduce functionalities that allow connecting image segmentation lines to cadastral boundaries regardless of their boundary likelihood.
Line 103-104: please add some more details about the aerial image. Which season was it captured in and how does that compare to the local agricultural cycle? Which phase of cropping are the fields likely to be in, what are the main crops or management forms?
We added the following:
The local agricultural practice consists mostly of smallholder farming. The crops appear to be in the beginning of an agricultural cycle as they do not cover the visible cover parcel boundaries.
Line 109, page 3: referring to a submitted paper for technical description of capture and processing should be avoided. Please provide a few sentences describing the technical basics here, and should the other paper be accepted during the review process of this paper, you can still add the citation and shorten the text. Otherwise you risk publishing this paper earlier than the other one and the reader might not be able to find the necessary information because the paper you cite does not exist (yet). Please consult with the technical staff or production editor about the journal policy on (the formatting of) citation of material in review.
We took out the reference to the submitted manuscript and added the technical basis in the text:
The UAV data in Rwanda have a GSD of 0.02 m and were captured with a rotary-wing Inspire 2 (DJI). The UAV data in Kenya have a GSD of 0.06 and were captured with a fixed-wing DT18 (Delair Tech).
Line 116: typo: "Ethiopia is use" should be "Ethiopia is used."
Changed.
Lines 237-238: "We used 10% of the balanced training data for validation" : this seems like a very small set of data used for validation. Please justify or explain.
We modified the text to:
Out of the 50% of balanced data used for training, we used 10% for validation. These data were not seen by the network, but used only to calculate loss and accuracy per epoch. These metrics and their curves looked most promising for VGG19 [25]. Then, we applied the trained network to the remaining 50% of testing data.
Lines 270-272: It seems that you left some text from the Latex template file in the final manuscript text. I assume this was not on purpose - please remove.
Removed.
Lines 300-308 "we filtered the cadastral reference dta for Ethiopia to contain visible parcels only etc": How much of the data was removed by this step? What do the 38% and the 71% refer to? Is this in pixels or parcels, or boundary segments, How much do you think it influences the final accuracy, including the less visible parcel boundaries? It would seem that the final accuracy you reach is strongly influenced by this step, which is not clearly documented. If you only keep the "easy cases" it would not be surprising that your accuracy is high. On the other hand, if you are comparing to hand digitized boundaries, they will also have errors where the parcel boundaries are not visible.
We modified the text to:
Indirect surveying, comprising of manual or automated delineation both rely on visible boundaries. Before comparing manual to automated delineation, we filtered the cadastral reference data for Ethiopia (Figure 2b) to contain visible parcels only. We kept only those parcels for which all boundary parts were visually demarcated. As in [4], we consider only fully closed polygons that are entirely visible in the image. From the original cadastral reference data, we kept 38% of all parcels for which all boundaries were visible. In [4], the portion of fully visible parcels has been reported to average around 71% of all cadastral parcels in rural Ethiopian areas. We can confirm 71% for parts of our study area that cover smallholder farms.
Line 320 "Figure 6Error! Reference source not found.d" : seem like this is a latex problem her
Changed.
Line 376 Discussion: discussion should quantitatively compare the accuracies and performance reached by the precursor of this method (described in Crommelinck, Höfle et al 2018) and the new method published here (even if the data used are not the same), adding in a short paragraph on this comparison. Also, the remaining sources of error and uncertainty should be discussed, ideally in a separate subsection. Please check the requirements for this section in the author guidelines.
We inserted a new section to the discussion to compare to our previous work, added headings for a clearer structure and emphasized on the limitations.
4.1 Working Hypothesis: Improving Boundary Mapping Approach
Compared to our previous workflow [21], we improved each of the three workflow steps. For image segmentation, we remove the previous need to reduce the image resolution for images larger than 1000 x 1000 pixels and we introduce a filtering step that allows to limit over-segmentation by reducing the number of segment lines by 80%. For boundary classification, we implement Convolutional Neural Networks (CNN) and thereby eliminate the previous need for Random Forest (RF) hand-crafted feature generation. For interactive delineation, we develop two additional delineation functionalities (‘Connect around selection’, ‘Connect lines’ endpoints’), we develop an attribute selection for the least-cost-path functionality (‘Connect along optimal path’) and redesign the GUI to be more intuitive. While we previously tested our approach on road outlines only, we now show advantages compared to manual delineation for cadastral mapping, which includes various object types. The number of clicks per 100 m compared to manual delineation was previously reduced by 76% and 86% when delineation roads from two UAV images. Now, we applied our approach to delineate 378 visible cadastral boundaries from UAV and aerial imagery of larger extents, while requiring on average 80% fewer clicks compared to manual delineation
4.2 Working Hypothesis: CNN vs. RF
By reformulating our problem to be solvable by a CNN, we have investigated integrating a more state-of-the-art approach in our previously proposed boundary delineation workflow [21]. A deep learning CNN was assumed to be superior to machine learning RF, as CNNs require no hand-crafted features and can be trained incrementally.
This starting hypothesis held true: even though pre-trained on images from computer vision, transfer-learning a CNN on remote sensing data provided more accurate predictions for boundary likelihoods compared to RF. Our successful integration reduces the effect of possibly meaningless or biased hand-crafted features and increases the degree of automation of our approach. However, when conducting the final workflow step, i.e., interactive delineation, we found that we seldom made use of the boundary likelihood. We reduced over-segmentation, due to post-processing the image segmentation. This, in combination with new interactive delineation functionalities, is more effective than manual delineation for regular-shaped parcels surrounded by visible boundaries. The delineation functionality that uses boundary likelihood is beneficial for long or curved boundaries, which was rare in our study areas.
4.3 Limitations & Future Work
When training a network to predict boundary likelihoods for visible object outlines, our training data based on cadastral reference are beneficial as it is available without further processing. The data have little bias, as no human annotator with domain knowledge is required [36]. However, the data could be improved: cadastral data contain invisible boundaries not detectable by MCG. To limit training data to visible boundaries would match better with what the network is expected to learn and increase achievable accuracy metrics. When deciding whether to use RF or CNN for boundary classification, one needs to balance feature extraction for RF [37] against training data preparation and computational requirements for CNN [18]. In cases of limited training data for CNN, our CNN-based boundary classification may be adopted by data augmentation and re-balancing class weights. One advantage of our RF-based boundary classification is that it contains a feature capturing 3D information from a Digital Surface Model (DSM), which we used in [21]. 3D information still needs to be included in the CNN-based boundary classification. Compared to computer vision, the amount and size of benchmark image data are marginal: existing benchmarks cover aerial data for urban object classification [38] and building extraction [39], satellite imagery for road extraction, building extraction and land cover classification [40], as well as satellite and aerial imagery for road extraction [41]. Such benchmarks in combination with open data initiatives for governmental cadastral data [42], aerial imagery [43], and crowdsourced labelling [44-46] may propel deep learning frameworks for cadastral boundary delineation, i.e., cadastral intelligence. Instead of using a VGG pre-trained on ImageNet, our approach could then be trained on diverse remote sensing and cadastral data, resulting in a possibly more effective and scalable network.
Despite the shown advances, automating cadastral boundary delineation is not at its end. Identifying areas in which a large portion of cadastral boundaries is visible and for which high-resolution remote sensing and up-to-date cadastral data are available in digital form still impedes methodological development. Future work could investigate the approach’s applicability for invisible boundaries, that are marked before UAV data capture, e.g., with paint or other temporary boundary markers. In this context, the degree to which the approach can support participatory mapping could also be investigated. Furthermore, research needs to be done on how to align innovative approaches with existing technical, social, legal, and institutional frameworks in land administration as addressed in [47-49]. We are currently pursuing this by developing documentation and testing material [50] that enables surveyors and policy makers in land administration to easily understand, test, and adapt our approach.
4.4 Comparison to Previous Studies
How we reformulated our problem to be solvable by a tile-based CNN has been similarly proposed in biomedical optics [51]. Fang et al. crop tiles centered on retinal boundary pixels and train a CNN to predict nine different boundary labels. Correspondingly labelled pixels are connected with a graph-based approach. To transfer the latter to our case, we may investigate whether connecting tiles of similar boundary likelihood can omit the need for an initial MCG image segmentation: by using Fully Convolutional Networks (FCN) [52] each pixel of the input image would be assigned a boundary likelihood, which can be connected using Ultrametric Contour Maps (UCM) [53] included in MCG as investigated in [54]. Connecting pixels of corresponding boundary likelihoods could also be realized by using MCG-based contour closure [55], line integral convolution [56], or template matching [57].
Alternatively, the topology of MCG lines can be used to sort out false boundary likelihoods before aggregating them per line: this could be realized by not shuffling training data and thus maintaining more context information per batch, or by using graph-based approaches such as active contour models [58] suggested for road detection [59,60], or region-growing models suggested for RF-based identification of linear vegetation [61].
Predicting the optimal MCG parameter k per image may also be achieved with CNNs. Depending on whether an area is, e.g., rural or urban, cadastral parcels vary in size and shape. Larger parcels demand less over-segmentation and a higher k. Similarly, our high-resolution UAV data required a higher k, i.e., 0.3 and 0.4 as compared to 0.1 for the aerial data. Challenges to be addressed are training with data from multiple sensors, varying parcel sizes in training, and automatically labelling data with the optimal segmentation parameter k.
In the discussion, you should discuss the implications of your findings not only in the context of technological development of boundary detection but also from the perspective of the user: will your advances really make a difference for the cause you state in the introduction ( recording land rights for security)? You might also want to mention the eventual relevance of this method for land parcel change detection in an EU agricultural subsidies monitoring context: in many countries the updating of the LPIS is still done by hand digitizing.
Please see comment above.
Also, we added agricultural monitoring in as further application field in the conclusion:
While our approach has been developed for cadastral mapping, it can also be used to delineate objects in other application fields, such as land use mapping, agricultural monitoring, topographical mapping, road tracking, or building extraction.
Line 421-456: Please see author guidelines here as well: "Conclusions should be self-contained, a reader should not have to read through the full paper to understand it". It would be essential that you quantitatively formulate the improvement in accuracy you reached compared to the previous findings.
We modified the conclusion accordingly:
5. Conlsuion
We have introduced a workflow that simplifies image-based delineation of visible boundaries to support the automated mapping of land tenure from various sources remote sensing imagery. In comparison to our previous work [21], the approach is now more automated and more accurate due to the integration of CNN deep learning compared to RF machine learning. For RF-derived boundary likelihoods, we obtained an accuracy of 41% and a precision of 49%. For CNN-derived boundary likelihoods, we obtained an accuracy of 52% and a precision of 76%. CNNs eliminate the need to generate hand-crafted features required for RF. Furthermore, our approach has proven to be less tiring and more effective compared to manual delineation due to the decreased over-segmentation and our new delineation functionalities. We limit over-segmentation by reducing the number of segment lines by 80% through filtering. Through the new delineation functionalities, the delineation effort per parcel requires 38% less time and 80% fewer clicks compared to manual delineation. The approach works on data from different sensors (aerial and UAV) of different resolutions (0.02-0.25 m). Advantages are strongest when delineating in rural areas due to the continuous visibility of monotonic boundaries. Manual delineation remains superior in cases where the boundary is not fully visible, i.e., covered by shadow or vegetation. While our approach has been developed for cadastral mapping, it can also be used to delineate objects in other application fields, such as land use mapping, agricultural monitoring, topographical mapping, road tracking, or building extraction.
Line 446-456: this section is more of a future outlook and should be a separate sub-section of the discussion instead of part of the conclusions.
We moved the section to the discussion (see above).
Reviewer 3 Report
Dear authors and editors,
This manuscript presents an improvement/refinement on several previous work of the authors. The study is well-described, documented, and presented.
Except for one typo error, I have no remarks whatsoever:
l.115-116: „In this study, an aerial image of..” à Delete. Repetition of l.103
I congratulate the authors with their nice work and manuscript.
Author Response
Dear reviewer,
thanks for having taken the time to review our submission and for providing your feedback.
We have updated our manuscript and corrected the issue you pointed out in lines 115-116.
Kind regards,
Sophie Crommelinck et al.